# Synergisms between Surfactants, Polymers, and Alcohols to Improve the Foamability of Mixed Systems

Luís Alves [1],*, Solange Magalhães [1], Cátia Esteves [2], Marco Sebastião [3] and Filipe Antunes [2]

1. University of Coimbra, CERES, Department of Chemical Engineering, Rua Silvio Lima, 3030-790 Coimbra, Portugal; solangemagalhaes@eq.uc.pt
2. University of Coimbra, CQC, Department of Chemistry, Rua Larga, 3004-535 Coimbra, Portugal; catia.esteves@uc.pt (C.E.); filipe.antunes@ci.uc.pt (F.A.)
3. Mistolin Company, Zona Industrial De Vagos, Lote 58, 3844-909 Vagos, Portugal; marco.sebastiao@mistolin.pt
* Correspondence: luisalves@ci.uc.pt

**Abstract:** In order to produce detergents with improved performance and good market acceptability, it is crucial to develop formulations with improved foamability and cleaning performance. The use of a delicate balance of surfactants and additives is an appealing strategy to obtain good results and enables a reduction in the amount of chemicals used in formulations. Mixtures of hydrophobically modified linear polymers and surfactants, as well as balanced mixtures with co-surfactants and/or hydrotropes, are the most effective parameters to control foamability and foam stability. In the present study, the effect of the addition of hydrophobically modified linear polymers, nonionic co-surfactants and hydrotropes, and their mixtures to anionic and zwitterionic surfactant aqueous solutions was evaluated. It was found that the presence of the hydrophobically modified polymer (HM-P) prevented the bubbles from bursting, resulting in better stability of the foam formed using zwitterionic surfactant solutions. Also, the surfactant packing was inferred to be relevant to obtaining foamability. Mixtures of surfactants, in the presence of a co-surfactant or hydrotrope led, tendentially, to an increase in the critical packing parameter (CPP), resulting in higher foam volumes and lower surface tension for most of the studied systems. Additionally, it was observed that the good cleaning efficiency of the developed surfactant formulations obtained a higher level of fat solubilization compared to a widely used brand of commercial dishwashing detergent.

**Keywords:** foamability; critical packing parameter; surface tension; hydrotropes; surfactants; adsorption

## 1. Introduction

Foam is a colloidal dispersion, in which a gas is dispersed in a continuous phase [1]. When air enters a surfactant solution, surfactant molecules become adsorbed at the air–water interface, and if the resulting surfactant monolayer stabilizes the air pocket, a bubble is formed, resulting in foam formation [2].

Once formed, foams can present different stabilities, characterized by the length of time that the foam can persist; it is possible to divide foams into unstable or transient foams (champagne bubbles are an example) and stable foams (for example, beer foam) [3,4].

Stable foams can present spherical bubbles but also foam cells, which are polyhedral and separated by flat liquid films. The more stable cell shape, in terms of minimizing surface free energy, is Kelvin's cell (tetrakaidecahedron), which consists of eight non-planar hexagon faces and six planar quadrilateral faces. It should be remembered, however, that foams contain a distribution of shapes, both spherical and polyhedral bubbles, depending on height and time [1,3], because their structure has a tendency to change under gravity, initially presenting a spherical shape, which has the tendency to change to a polyhedral form when the bubbles become dry [5].

The foam stability depends on some factors such as surface tension, surface rheological properties, surface forces, and film elasticity [6–8], with the last being considered the principal factor of foam stability [6,9].

Surface tension is related to the free energy of coalescence: when the surface tension decreases, the coalescence also decreases. However, other phenomena, like flotation, can contribute to foam instability and even to low surface tension values [10].

The rupture of foam films, and hence, the decrease in foam stability, is also affected by the surface forces, also called disjoining pressure, between the air–water interfaces [11]. Disjoining pressure is the pressure caused by the attractive forces between the two interfaces. In general, a positive (repulsive) disjoining pressure slows the process of rupture. However, at low surfactant concentrations, disjoining pressure cannot explain foam stability, which suggests there are other important factors [9].

Foam stability can be measured by the length of time the foam persists without being destroyed, which depends on the lifetime of the internal films separating the foam cells, i.e., film elasticity [12]. Film elasticity can be defined as the capacity to restore the initial state after a deformation. When a liquid film breaks up, foam cells become larger because of the coalescence of neighboring cells [6].

Thermodynamically, foams are unstable. For this reason, the kinetic stabilization of foams is required. Because of the synergistic effect with low molecular weight surfactants [13], polymers have been used as additives in foaming solutions to stabilize the foam, since polymers can also adsorb at the interfaces [11,14].

Polymer–surfactant complexes are formed in bulk and on the surface of the mixed solutions [2,15,16]. To control the processes of foam formation and stabilization, it is essential to have a good understanding of the involved interactions. Foam stability may increase in the presence of polymer–surfactant mixtures because the complexes formed are trapped in foam films and reduce drainage by increasing the viscosity of the solution [2].

Also, the surfactant molecules' arrangement at the air–water interface plays a crucial role in the stability of the foams [2,17]. As the critical packing parameter (CPP) of the surfactant increases, surfactants pack closer at the interface and higher concentrations of surfactant molecules adsorb at the monolayer, giving good strength, increased elasticity, and viscosity to the foam lamellae, leading to better foamability and appropriate foam stability [18]. Indeed, if foamability and foam stability are only governed by the surfactant packing at the interface, the continued increase in the CPP would be favorable to obtaining high foam volumes with enhanced stability [2]. However, other phenomena are involved, and foams are destroyed by coalescence due to the formation of holes in the boundary-thin liquid film of the foam cells [19]. Holes present a very large curvature; for that reason, the formation of holes is easier in surfactant systems with a high CPP than in low-CPP systems, because of the energetic penalty of forming large curvatures in low-CPP systems [20]. Consequently, good foamability and foam stability should be obtained for surfactants or surfactant systems with a low CPP. Therefore, a delicate CPP balance is necessary to obtain the maximum foam volume as well as good foam stability.

In the literature, studies on mixtures of polymers and surfactants [21–27], surfactant mixtures [28,29], combinations of silica nanoparticles [30–32], or alumina nanoparticles [33] with surfactants with regard to foam stability have been completed. Synergistic effects were seen between mixtures of cationic surfactants and cationic polymers, cationic surfactants and nonionic polymers, and furthermore, between anionic surfactants and nonionic polymers, considering foam stability and foam ability. For example, Deng et al. reported an enhancement in foamability and foam stability induced by interactions between a hyperbranched exopolysaccharide and a zwitterionic surfactant dodecyl sulfobetaine [23]. Positive effects on foamability were also reported by Momin and Yeole due to the formation of nonionic polymer–anionic surfactant complexes in aqueous solutions [16]. On the other hand, Wang et al. addressed a study demonstrating the foam stability gain obtained for "catanionic" surfactant mixtures of sodium dodecyl sulfate (SDS) and dodecyl trimethyl ammonium bromide ($C_{12}TAB$) [28]. Similarly, Almobarky et al. showed that the use of mix-

tures of two anionic surfactants leads to the enhancement of foam stability, with potential application in oil recovery [29]. Also, with application in oil recovery, Babamahmoudi and Riahi used silica nanoparticles to improve foam stability in the presence of crude oil [30]. Likewise, Yang et al. demonstrated the foam-stabilizing effect of alumina nanoparticles in sodium cumenesulfonate aqueous solutions, for application in oil recovery [33]. On the contrary, the combination of a cationic polymer and anionic surfactant harmed the foam stability and foam boost [2]. It was also demonstrated that a higher foaming power is not necessarily correlated with higher stability. Ali et al. investigated foam stabilization using silica nanoparticles, attributing this stabilization to surfactants migrating and remaining at the oil–water interface, induced by the presence of the nanoparticles, thereby forming a robust layer that prevents contaminant coalescence [34].

In the present study, the effect of the addition of hydrophobically modified linear polymers, nonionic co-surfactants and hydrotropes, and their mixtures, on the foam capacity of an anionic surfactant (alcohols, C12-14, ethoxylated, sulfates, sodium salts (2 EO)—$SLE_2S$) and zwitterionic surfactant (cocamidopropyl betaine—CAPB) solution were evaluated through its foamability (foam formation and foam stability) and tensiometry. In the literature, it is possible to find works dealing with foam stabilization, mainly focused on the use of one additive, such as a polymer or nanoparticles. However, to the best of our knowledge, no study applying different additives able to boost foamability and foam stability, using different types of surfactants, has been performed. The present work intends to contribute to the development of surfactant formulations with good foamability and improved cleaning performance, taking into account their versatility across various sectors, including but not limited to the detergent, cosmetic, and other industries.

## 2. Materials and Methods

### 2.1. Materials

Propanaminium, 3-amino-N-(carboxymethyl)-N,N-dimethyl, N-coco acyl derivatives, hydroxides, inner salts (betaine) (35% ($w/w$) aqueous solution), oxirane, 2-methyl-, polymer with oxirane, mono(2-propylheptyl) ether (lut) (>99% ($w/w$) active surfactant), and alcohols, C12-14, ethoxylated, sulfates, sodium salts (2 EO) ($SLE_2S$) (70% ($w/w$) aqueous solution) were supplied by BASF GmbH. The surfactant structures are depicted in Figure 1. Hydrophobically modified alkali-soluble acrylic polymer (HM-P) and dipropylene glycol n-butyl-ether (Dpnb) were obtained from Dow Chemical. The reagents were used without any further purification.

**Figure 1.** Chemical structures of the surfactants used: (**a**) oxirane, 2-methyl-, polymer with oxirane, mono(2-propylheptyl) ether, where $m = 1$, $n = 6$ (lut); (**b**) alcohols, C12-14, ethoxylated, sulfates, sodium salts (2 EO) (SLE2S); (**c**) propanaminium, 3-amino-N-(carboxymethyl)-N,N-dimethyl-, N-coco acyl derivatives, hydroxides, inner salts (betaine).

## 2.2. Methods

### 2.2.1. Sample Preparation

Solutions were prepared using Mili-Q water with a conductivity of 18.2 M$\Omega$·cm$^{-1}$. Solutions to evaluate the formed volume of foam were prepared to contain 7% (*w/v*) (ca. 0.204 M of betaine and 0.186 M of SLE$_2$S) surfactant (stock solutions). The selected additives (HM-P, lut, and Dpnb) were added to a stock solution in order to evaluate their effect on the formed volume of foam and its stability. HM-P was added at a concentration of 0.014% (*w/v*), lut was added at a concentration of 0.116% (*w/v*) (ca. 2.37 mM), and Dpnb was used at 0.35% (*w/v*) (ca. 18.39 mM) concentration. The pH of all solutions was adjusted to 7.0.

To study the surface tension, stock solutions containing 3.5 g·L$^{-1}$ of surfactant in Mili-Q water were prepared for each surfactant separately.

### 2.2.2. Foaming Capacity Measurement

The foam was analyzed following the ASTM D-1173-53 (Standard Test Method for Foaming Properties of Surface-Active Agents), analyzing the volume of foam at t = 0 min, t = 5 min, t = 10 min, and t = 30 min (Figure 2).

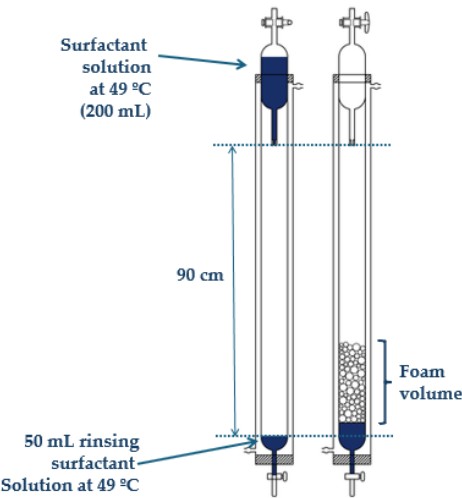

**Figure 2.** Standard test apparatus for foaming properties of surface-active agents (ASTM D-1173-53).

The apparatus and temperature used were described in the previously mentioned standard. Briefly, the pipet was filled with 200 mL of solution containing 1.0 g surfactant·L$^{-1}$ of the different surfactants (with or without additives) and 50 mL of solution was used to rinse the walls of the receiver cylinder; then, the pipet was refilled to the 200 mL mark, placed in position at the top of the receiver cylinder, and the stopcock was opened. After all the solution had run out of the pipet, the foam volume produced was measured through the reading of the foam height, and the obtained height was used to calculate the foam volume (Vfoam = πR2h, where h is the foam height). The cylinder used had a 5 cm internal diameter. All the tests were carried out at 49 °C and 5 repetitions were performed for each formulation. The foam stability was analyzed by measuring the volume at different times (from *t* = 0 min to *t* = 30 min) and by observation using a digital high-resolution camera (Moticam 2.0) and the Motic Images Advanced 3.2 software (MoticEurope, S.L.U., Barcelona, Spain) at the different foam ageing times previously described.

### 2.2.3. Surface Tension Measurement

Surface tension measurements were carried out in an Attension Sigma 702 (Biolin Scientific, Gothenburg, Sweden) at 25 °C, using the Du Noüy ring method based on force measurements. The platinum ring was thoroughly cleaned and dried before each measurement. Each measurement was repeated up to six times to check for reproducibility.

Small volumes (20 μL) of the surfactant solution (containing 3.5 g·L$^{-1}$ surfactant) were successively added to 10 mL Mili-Q water and surface tension was measured for the different surfactant concentrations.

2.2.4. Fat Solubilization Capacity

The oil/fat solubilization capacity of the different formulations was evaluated following a procedure described by Rao et McClements [35], with the necessary adaptations. Aqueous solutions containing 1.0 g·L$^{-1}$ of the different surfactants (with or without additives) were prepared by weighing the necessary amount of the different formulations and dissolving them in distilled water. A 5 mL solution of the different surfactants and 1 mL of vegetable oil (used as the standard fat/oil) were added into a container and then blended together using a high-speed stirrer (1000 rpm) for 1 min at room temperature. The resultant emulsions were left to rest for 10 min, and then the amount of oil on the top of the aqueous solution was measured. The solubilization efficiency was determined by the difference between the initial amount of oil added (1 mL) and the oil "out" of the aqueous phase, for a fixed surfactant concentration (1.0 g·L$^{-1}$).

**3. Results and Discussion**

Foamability and foam stability are important parameters in many applications such as detergents (for example, dishwashing detergents), personal care, and cosmetics. In the present study, the effect of three additives of different natures and their mixtures on the foaming capacity of two surfactants with broad industrial use (SLE$_2$S and cocamidopropyl betaine) was evaluated. The foam volumes generated by the different solutions of surfactants and surfactant/additive mixtures, using the standard method of Ross Miles (ASTM D-1173-53), are presented in Figure 3.

It was found that additives have different effects on foaming and foam stability, depending on the surfactant nature. The addition of HM-P to the anionic surfactant SLE$_2$S resulted in an increase in the foam volume at *t* = 0 min; however, the foam collapsed faster than in the absence of hydrophobically modified polymer. On the contrary, when added to the zwitterionic surfactant, it led to a significant foam volume decrease and a slight increase in foam stability, as can be seen in Figure 3. The reduction in foam volume of the betaine–HM-P system can be attributed to the formation of surfactant–polymer complexes in bulk, leading to a reduction in the number of surfactant molecules available at the air–water interface [2].

On the other hand, the addition of a branched nonionic surfactant increased the foam volume generated by the aqueous solution of betaine, compared with the original solution, and decreased the foam volume of the anionic SLE$_2$S. Similar results were obtained by the addition of a co-solvent (Dpnb), with a small difference in foam boost obtained for the betaine solution, with the foam volume after thirty minutes being higher than the foam volume of the betaine solution at *t* = 0 min. The combination of the hydrophobically modified polymer and the nonionic surfactant with SLE$_2$S led to an increase in the foam stability but lower foam volume; on the other hand, the betaine solution presented better foam stability with the addition of the mixture of HM-P and lut, with the foam volume also being higher than for the solution containing only betaine. Previous studies have shown that the interaction of HM-P with anionic surfactants, such as SLE$_n$S, is favored compared to an HM-P–betaine interaction, which could explain the results obtained for the decrease in foam volume with the presence of HM-P in betaine solutions [36,37]. Ostwald ripening is one of the mechanisms of foam and emulsion instability, driven by a spontaneous process that occurs because larger particles are energetically more favored than smaller particles. The addition of HM-P changes the rheology of the solution film due to the complexes formed between the hydrophobically modified polymer and surfactant, leading to an increase in the viscosity and stability of the solution film [38,39]. Also, to hinder Ostwald ripening, the use of surface active polymers such as hydrophobically modified polymers is usually very useful because these species adsorb irreversibly at the air–water

interface and the polymer surface concentration is the same on small droplets as it is on large droplets, reducing the pressure in the smaller droplets and hampering the Ostwald ripening process [20]. As consequence, the foam stability is increased, as can be seen in the case of betaine/HM-P mixed solutions. For the $SLE_2S$ solutions, the addition of HM-P did not increase the stability. As previously suggested, different factors can contribute to foam instability, and we can conclude that Ostwald ripening should not be the dominant one in the $SLE_2S$ case.

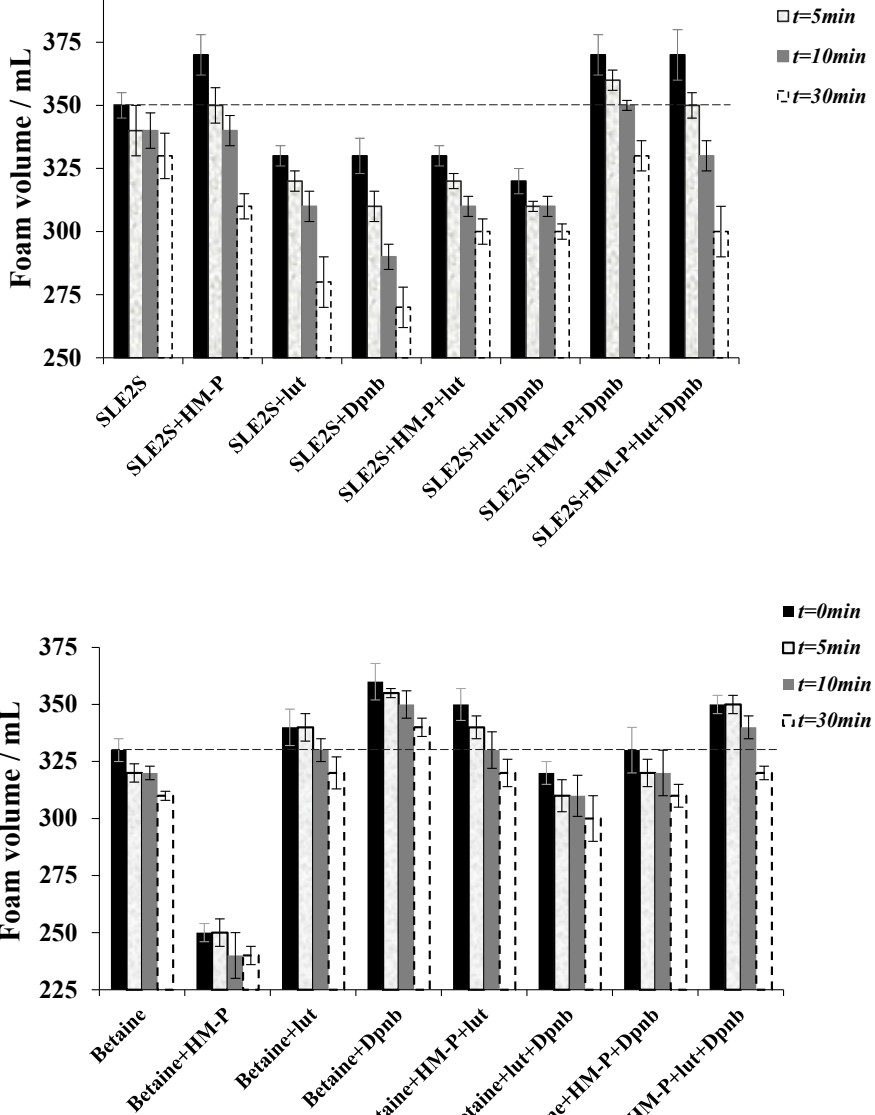

**Figure 3.** Foam volume as a function of time for single surfactant system aqueous solutions ($SLE_2S$—**top**; betaine—**bottom**) and mixtures of these surfactants with HM-polymer, co-surfactant, and co-solvent systems. The concentration used was kept constant with a value of $0.7 \, g \cdot L^{-1}$ and a solution pH of 7.0, and the temperature used was 49 °C.

The shape and size of the foam cells formed by surfactant aqueous solutions in the absence and presence of HM-P were evaluated by visualization on a high-resolution camera (Figure 4).

## SLE$_2$S aqueous solution

## SLE$_2$S + HM-P aqueous solution

**Figure 4.** Foam observed on a digital high-resolution camera at $t = 0$ min and $t = 30$ min, at 25 °C, for surfactant (SLE$_2$S) aqueous solutions with and without HM-P. The scale bar represents 5 mm.

As can be seen, at $t = 0$ min the bubble size was smaller than that observed at $t = 30$ min, as expected. It was observed that the bubble size and shape are dependent on the polymer incorporation in the formulation. At $t = 0$ min, for the same surfactant concentration, the solutions presented bubbles of smaller size when the polymer was incorporated and at $t = 30$ min the polymer effect was even more pronounced, with the system betaine/HM-P being the one presenting lower coalescence. It is also possible to observe that the shape of the cells was not spherical but polyhedral, separated by flat liquid films, indicating good foam stability [40,41] with a very small decrease in foam volume, ca. 10 mL for the betaine/HM-P system, after 30 min.

On the other hand, the addition of a branched tail surfactant (lut) can change the packing of the surfactant molecules and consequently lead to variations in some important parameters, such as surface tension [41]; as a result, differences in foamability and critical micelle concentration (cmc) are expected. In Figure 5, the results obtained for the cmc of the different formulations of SLE$_2$S (black columns) and betaine (grey columns) are presented.

As can be seen, the addition of HM-P or lut resulted in a decrease in the cmc values of SLE$_2$S and betaine surfactants. For the SLE$_2$S case, the addition of the HM-P shifted the micelle formation from 0.073 g·L$^{-1}$ to 0.052 g·L$^{-1}$. The obtained cmc value for the solution containing SLE$_2$S alone was lower than the reported value, about ca. four times less than the literature value [42]; this can possibly be explained by the presence of impurities, as the surfactants were commercial solutions. The addition of the HM-P led to the formation of micelles at a lower surfactant concentration, the denominated critical aggregation concentration (cac), driven by a hydrophobic attraction between the polymer and the surfactant molecules. Such interactions are particularly strong for grafted copolymers, with long hydrophobic groups grafted onto a hydrophilic polymer backbone, termed block copolymers [43].

It is known that mixtures of surfactants give rise to lower cmc values compared with single surfactant solutions [44]. Our results show that the addition of lut to SLE$_2$S and betaine solutions resulted in a significant cmc value decrease. The effect was more pronounced for the SLE$_2$S case due to the surfactant nature; both SLE$_2$S and betaine are ionic surfactants, but the charge density of the anionic one is higher than the zwitterionic; as result the cmc of betaine was lower than SLE$_2$S, due to the higher charge repulsion of the

ionic surfactant head. The addition of a nonionic co-surfactant to ionic surfactant solutions led to a decrease in the repulsion of charged surfactant heads and entropic penalty due to the counterions [44]. A similar result was obtained with the addition of Dpnb to the surfactant solutions, but with a lower impact. This lower efficiency can be attributed to two different factors: Dpnb is a nonionic hydrotrope and can reduce the surfactant head charge repulsion by the formation of mixed surfactant/hydrotrope micelles; however, the chain length of Dpnb is very small compared with the surfactant tail and as a consequence possesses lower surface activity. The second factor is related to surfactant solubility, which is increased by the presence of the hydrotrope [45,46]; as a consequence, the cmc values increase. The balance between these two factors resulted in a slight decrease in the cmc of $SLE_2S$.

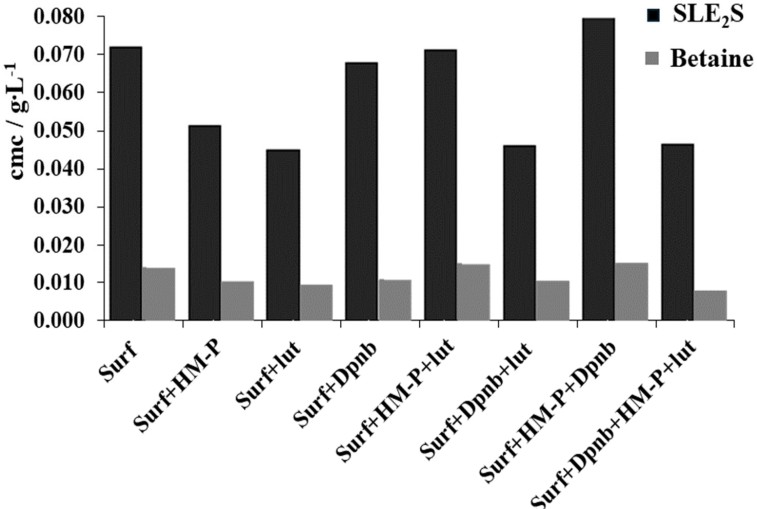

**Figure 5.** Critical micelle concentration of individual surfactants ($SLE_2S$—black columns and betaine—grey columns) and their mixtures with additives (hydrophobically modified polymer (HM-P), branched nonionic surfactant (lut), and a co-solvent (Dpnb)). All measurements were performed at 25 °C and pH 7.0.

The increase in surfactant solubility was not favorable if combined with the presence of HM-P, which resulted in a cmc value higher than the isolated surfactants. Nevertheless, the system presenting the lower cmc values contained a complex mixture of surfactant, polymer, co-surfactant, and hydrotrope. These systems had values of cmc ca. $0.045 \ g \cdot L^{-1}$ (for $SLE_2S$) and $0.008 \ g \cdot L^{-1}$ (for betaine), which is roughly half of the cmc value of the surfactants alone.

Surfactants are surface active species and consequently are very effective in changing solution physical properties, as is the case with the surface tension. Figure 6 shows the minimum surface tension values obtained for the different surfactant solutions, isolated and in combination with other compounds.

The surface tension of the surfactant solutions alone was lower for betaine than for $SLE_2S$. Anionic or zwitterionic surfactants combined with lut and Dpnb led to a reduction in the surface tension of the surfactant solutions. This reduction was more pronounced for mixtures of betaine and lut, reducing the surface tension to $29.4 \ mN \cdot m^{-1}$, compared with $30.4 \ mN \cdot m^{-1}$ for betaine alone. However, the $SLE_2S$ solution also presented a reduction in surface tension values when mixed with lut or Dpnb, compared with the single surfactant solution. This reduction in surface tension can be attributed to better packing of the surfactants at the air–water interface, induced by the presence of the co-surfactant or the hydrotrope, thus attaining greater cohesion. The presence of the HM-P also affected the surface tension of the surfactant solutions, and this phenomenon was more pronounced in the case of betaine, where a slight increase in the surface tension value was observed; this increase can be attributed to the formation of surfactant–polymer complexes in bulk, which reduces the amount of surfactant at the interfaces, as previously discussed.

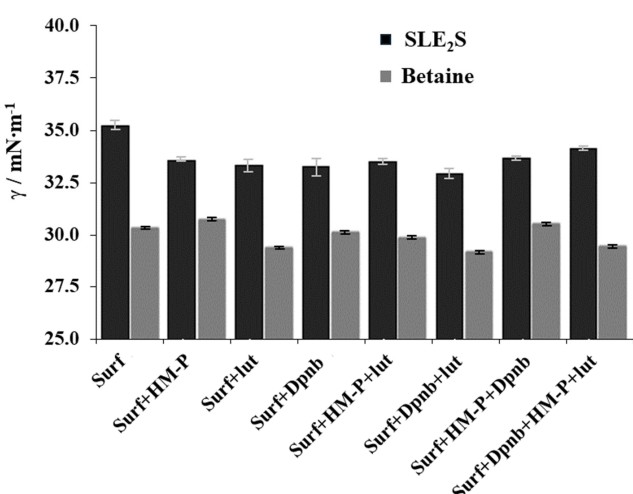

**Figure 6.** Surface tension of aqueous solutions of individual surfactants (SLE$_2$S—black columns and betaine—grey columns) and mixtures of surfactants and additives at a constant concentration (0.7 g·L$^{-1}$) at 25 °C and pH 7.0.

The changes in surface tension of a solution are directly related to the adsorption of solutes, e.g., surfactants, to the air–solvent interface. The Gibbs equation for nonionic surfactants is

$$\Gamma^{(1)} = -\frac{1}{RT}\frac{d\gamma}{dlna} \tag{1}$$

and for ionic surfactants is

$$\Gamma^{(1)} = -\frac{1}{2RT}\frac{d\gamma}{dlna} \tag{2}$$

where a is the activity of the solute in bulk. This gives us the relationship between the adsorption of surfactants and the surface tension of the solution; for surfactant concentrations below the cmc, the amount of surfactant in bulk is particularly reduced, and the surfactant activity can be replaced by the surfactant concentration. If we assume that surfactants form a monolayer at the interface, it is possible to calculate the area occupied by a single surfactant molecule, as the quantity of surfactant molecules adsorbed is inversely proportional to the area occupied by each molecule [47]. By plotting surface tension as function of surfactant concentration, the slope below the cmc value is indicative of the adsorption of the surfactants to the solute interface; two examples are given in Figure 7. An increase in the absolute value of the slope is indicative of improved surfactant adsorption to the interface, according to the previously mentioned equations. Table 1 presents the obtained results for adsorption, the area occupied by single surfactant molecules, and the CPP for the different system.

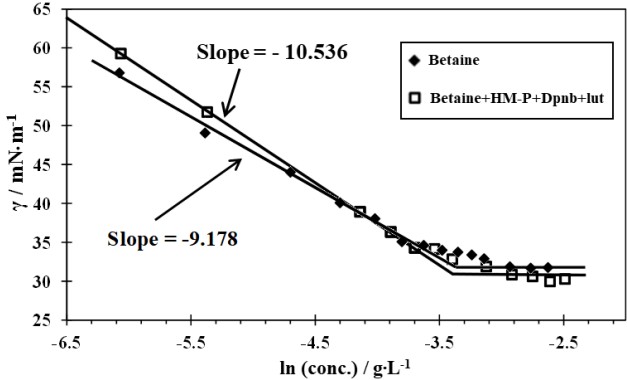

**Figure 7.** Surface tension of aqueous solutions of an individual surfactant (betaine) and mixtures of surfactants and additives at a constant concentration (0.7 g·L$^{-1}$) at 25 °C and pH 7.0.

**Table 1.** Surface excess ($\Gamma$), molecular cross-sectional area (A), and calculated CPP values for the different surfactant solutions at 25 °C and pH 7.0.

| Surfactant System | | $\Gamma/10^{-10}$ mol·cm$^{-2}$ | A/Å Molecule$^{-1}$ | CPP |
|---|---|---|---|---|
| **SLE$_2$S** | | 1.30 | 127.45 | 0.16 |
| | + HM-P | 1.50 | 111.07 | 0.19 |
| | + Lut | 1.81 | 91.53 | 0.23 |
| | + Dpnb | 1.79 | 92.88 | 0.23 |
| | + HM-P + lut | 1.45 | 114.35 | 0.18 |
| | + lut + Dpnb | 2.09 | 79.30 | 0.26 |
| | + HM-P + Dpnb | 1.78 | 93.22 | 0.22 |
| | + HM-P + Dpnb + lut | 1.61 | 102.91 | 0.20 |
| **Betaine** | | 3.50 | 47.43 | 0.44 |
| | + HM-P | 4.28 | 38.77 | 0.54 |
| | + Lut | 4.03 | 41.25 | 0.51 |
| | + Dpnb | 4.70 | 35.30 | 0.59 |
| | + HM-P + lut | 4.26 | 38.95 | 0.54 |
| | + lut + Dpnb | 4.94 | 33.62 | 0.62 |
| | + HM-P + Dpnb | 4.01 | 41.40 | 0.51 |
| | + HM-P + Dpnb + lut | 4.29 | 38.73 | 0.54 |

As can be seen in Table 1, mixed solutions of two different surfactants or surfactants and additives promote adsorption at the interfaces. From the adsorption results, it is possible to calculate the CPP of the surfactants in the different systems. The critical packing parameter can be calculated using Equation (3):

$$\text{CPP} = \frac{v}{A \times l} \tag{3}$$

where $v$ is the hydrocarbon chain volume (in nm$^3$), assuming an incompressible fluid, and can be calculated using the following approach: $v \approx (27.4 + 26.9n) \times 10^{-3}$, where n is the number of carbon atoms in the surfactant chain; $A$ is the optimal headgroup area (obtained from adsorption results); and $l$ is the maximum effective length of the hydrophobic chain (in nm), corresponding to a semiempirical parameter known as the critical chain length ($l \leq l_{\max} \approx (0.154 + 0.1265n)$, where $n$ is the number of carbon atoms in the surfactant chain) [48]. The values obtained for the CPPs of the surfactants in the different systems are in the range of 0.16 to 0.26 for SLE$_2$S and 0.44 to 0.62 for betaine. For the SLE$_2$S systems, the results show that this surfactant tends to form cylindrical micelles, but for the system containing SLE$_2$S + lut + Dpnb, the CPP value (0.26) was close to the limit value of cylindrical micelles (0.33) [30]. On the other hand, the value of the CPP for betaine in aqueous solution was in the range of cylindrical micelles (0.33 < CPP < 0.5). Samples containing betaine mixed with additives presented values of CPPs in the range of flexible bilayer phases (0.5 < CPP < 1) [49]. This increase in the CPP, resulting in a better packing of the surfactants, resulted in higher volumes of foam, except for the systems containing betaine with HM-P and betaine + lut + Dpnb.

The use of complex mixtures containing anionic or zwitterionic surfactants in combination with adequate amounts of specific additives leads to higher foam volumes, as well as lower cmc and surface tension values. Additionally, these mixed systems are expected to have high cleaning efficiency. It was found that the solubilization capacity of the different complex mixtures was clearly superior to the isolated surfactant solutions. For example, the mixture containing anionic surfactant, HM-P, lut, and Dpnb was able to solubilize 0.3 mL of vegetable oil in 5 mL of an aqueous solution containing 1.0 g·L$^{-1}$ of surfactant (with additives). Conversely, the same surfactant (isolated) only was capable of solubilizing

0.11 mL of oil, and with the addition of HM-P, the solubilization value slightly increased to 0.12 mL. Similar results were obtained for betaine. It is also important to note that the result obtained for our complex mixture (surfactant + HM-P + lut + Dpnb) was approximately twice the result obtained for a commercial dishwashing detergent.

An illustrative schema is presented in Figure 8, presenting the main findings.

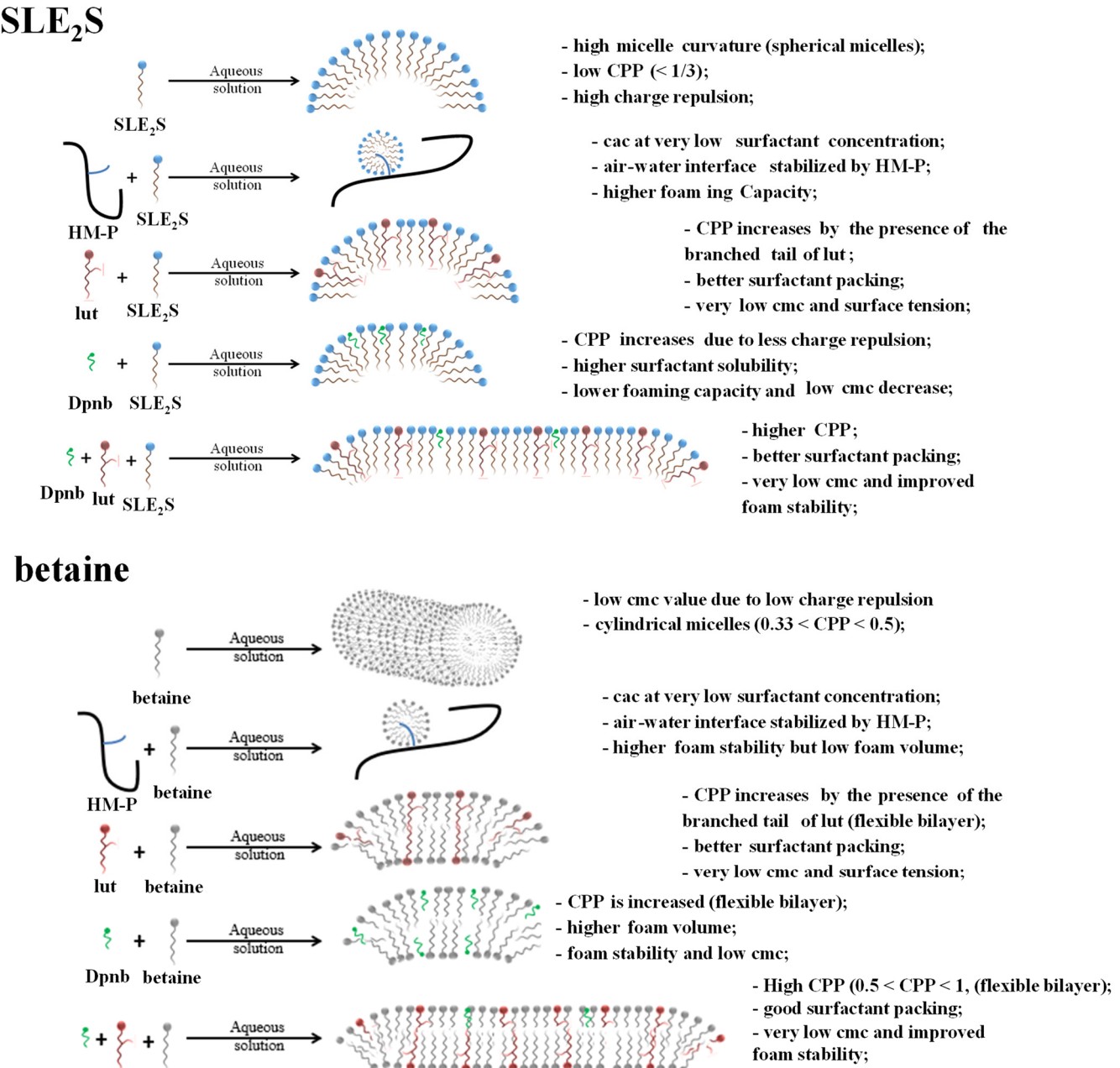

**Figure 8.** Schematic illustration of the effects of the different additives on the properties of SLE$_2$S and betaine aqueous solutions.

### 4. Conclusions

The foaming capacity of formulations is an important parameter for obtaining products with good acceptability in the market. Also, strategies that enable the use of smaller amounts of harmful chemicals in formulations represent a huge effort to reduce environmental impact and hazard to humans. In the present study, different additives were used to stabilize foams and generate higher foam volumes. It was found that the way the surfac-

tants pack plays a crucial role in the volume and stability of the generated foams. High CPP values, enabling the surfactants to pack closer together at the air–solvent interface, lead to liquid films with better strength, enhanced elasticity and viscosity, and thus better foamability and proper foam stability. An increase in the CPP was obtained for mixtures of surfactants with nonionic co-surfactants or hydrotropes, resulting, in the case of the zwitterionic surfactant, in better foamability. The values of critical packing parameters for betaine and additive mixtures were in the range of flexible bilayer phases (0.5 < CPP < 1). Flexible bilayer phases are very efficient at encapsulating hydrophobic compounds, predicting good cleaning efficiency for formulations based on betaine and additives.

Furthermore, the addition of lut and Dpnb to ionic surfactant solutions resulted in a decrease in the critical micelle concentration, and when added to a zwitterionic surfactant, a reduction in surface tension was also obtained. The use of complex mixtures containing anionic or zwitterionic surfactants in combination with adequate amounts of HM-P, lut, and Dpnb led to higher foam volumes, as well as lower cmc and surface tension values. The results obtained in the present work illustrate the synergistic effects of mixtures of surfactants with specific additives, studying the effect of the individual additives separately. It was shown that this synergic combination is able to potentiate foamability, and positively impacts cleaning performance. These mixed systems are adequate to be applied in high-performance formulations where foam and a high cleaning efficiency play major roles.

**Author Contributions:** L.A.: Conceptualization, Investigation, Methodology, Writing—original draft, Writing—review and editing, Project management. S.M.: Investigation, Methodology, Writing—review and editing. C.E.: Investigation, Methodology, Writing—review and editing. M.S.: Writing—review and editing. F.A.: Funding acquisition, Writing—review and editing. All authors have read and agreed to the published version of the manuscript.

**Funding:** The authors are thankful to the Portugal 2020 Program (Project SUGARDWASH), reference POCI-01-0247-FEDER-010954, for the financial support. The CQC, supported by FCT projects UIDB/00313/2020 and UIDP/00313/2020, is also acknowledged. CERES, supported by the Portuguese Foundation for Science and Technology (FCT) through the Strategic Research Centre Project UIDB/00102/2020 (https://doi.org/10.54499/UIDB/00102/2020), is also acknowledged. Luís Alves acknowledges FCT for the financial support through the researcher grant 2021.00399.CEECIND (https://doi.org/10.54499/2021.00399.CEECIND/CP1656/CT0025).

**Institutional Review Board Statement:** Not applicable.

**Informed Consent Statement:** Not applicable.

**Data Availability Statement:** Data are contained within the article.

**Conflicts of Interest:** Author Marco Sebastião is employed by the company Mistolin Company. The remaining authors declare that the research was conducted in the absence of any commercial or financial relationships that could be construed as a potential conflict of interest.

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
