# Peer review of "Synergisms between Surfactants, Polymers, and Alcohols to Improve the Foamability of Mixed Systems"

_2571-8800, doi:10.3390/j7020010_

Round 1
Reviewer 1 Report
Comments and Suggestions for Authors
Manuscript ID: J-2956966
Synergisms between surfactants, polymers and alcohols to improve the foamability of mixed systems.
Luís Alves, Solange Magalhães, Cátia Esteves, Marco Sebastião, Filipe Antunes
In the presented study, the effect of the addition of hydrophobically modified linear polymers, non-ionic co-surfactant and hydrotropes, and its mixtures, on the foam capacity of an anionic surfactant (SLE2S) and a zwitterionic surfactant (CAPB) solutions were evaluated through its foamability and tensiometry.
During the work, only two fairly routine methods were used. A not very large block of experimental data was obtained. Nevertheless, the problem to be solved may be of interest to the scientific community and readers of the journal.
However there are numerous points that need to be revised before this manuscript can be recommended for publication.
The introduction should provide a rationale for choosing these particular research objects. It is also necessary to emphasize the relevance of the work by analyzing existing achievements, taking into account publications in recent years (the list of cited literature does not include articles published over the past 5 years).
The authors claim that the resulting compositions have good cleaning properties. This should be supported by some data. For example, the solubilization effect of the created compositions in relation to fats or other contaminants may be demonstrated.
The manuscript should contain not only a Results section, but also a Discussion one.
The illustrative material can be presented more compactly. For example, I recommend combining surface tension isotherms (Figure 6) of aqueous solutions of individual surfactants and mixtures of surfactants and additives on one graph. (By the way, authors should clarify the label of the x-axis in Figure 6: “ln” or “log”?). Figures 4 and 5 may be delete and the data presented in them may be added as additional columns in Table 2.
The illustrative schema resuming the main findings (Figure 7) must be redone. In the presented version, the upper part of the picture (SLE2S) does not differ from the lower one (betaine) and this does not agree with the presented conclusions (text on the right).
Taking into account the above, I believe that manuscript needs major revision before it can be recommended for publication.
Author Response
Reviewer #1
Manuscript ID: J-2956966
Synergisms between surfactants, polymers and alcohols to improve the foamability of mixed systems.
Luís Alves, Solange Magalhães, Cátia Esteves, Marco Sebastião, Filipe Antunes
In the presented study, the effect of the addition of hydrophobically modified linear polymers, non-ionic co-surfactant and hydrotropes, and its mixtures, on the foam capacity of an anionic surfactant (SLE2S) and a zwitterionic surfactant (CAPB) solutions were evaluated through its foamability and tensiometry.
During the work, only two fairly routine methods were used. A not very large block of experimental data was obtained. Nevertheless, the problem to be solved may be of interest to the scientific community and readers of the journal.
However there are numerous points that need to be revised before this manuscript can be recommended for publication.
The introduction should provide a rationale for choosing these particular research objects. It is also necessary to emphasize the relevance of the work by analyzing existing achievements, taking into account publications in recent years (the list of cited literature does not include articles published over the past 5 years).
Reply: We thank the reviewer’s comments and suggestions. We agree with the reviewer, and we updated the literature review of the introduction, which now includes recent references.
The authors claim that the resulting compositions have good cleaning properties. This should be supported by some data. For example, the solubilization effect of the created compositions in relation to fats or other contaminants may be demonstrated.
Reply: We agree with the reviewer. Solubilization tests were performed for the different formulations and a brief discussion was included in the manuscript. For example, our formulation containing surfactants, HM-P, lut and DPnB was able to solubilize 0.3 mL in 5 mL of an aqueous solution containing 1g/L surfactant, approximately twice the fat solubilized by a commercial dishwashing detergent at the same concentration. The procedure used to study fat solubilization was also added to the materials and methods section.
The manuscript should contain not only a Results section, but also a Discussion one.
Reply: The discussion section is combined with the results section. We rewrite the heading of the section to “Results and Discussion”.
The illustrative material can be presented more compactly. For example, I recommend combining surface tension isotherms (Figure 6) of aqueous solutions of individual surfactants and mixtures of surfactants and additives on one graph. (By the way, authors should clarify the label of the x-axis in Figure 6: “ln” or “log”?). Figures 4 and 5 may be delete and the data presented in them may be added as additional columns in Table 2.
Reply: We thank the reviewer's suggestion. Following the suggestion the data contained in Figure 6 was compacted in a single graph. The label of the x-axis was clarified (ln). About Figures 4 and 5 we believe that the way these results are presented is more intuitive and easier to follow than in a table. However, and considering the reviewer's suggestion, we merged the data from Figure 4 into a single graph, as well as the data contained in Figure 5.
The illustrative schema resuming the main findings (Figure 7) must be redone. In the presented version, the upper part of the picture (SLE2S) does not differ from the lower one (betaine) and this does not agree with the presented conclusions (text on the right).
Reply: We have redone the schema of Figure 7 to reflect the findings of the work

Reviewer 2 Report
Comments and Suggestions for Authors
Overall this paper is well structured and has the clean logic and necessary depth to be suitable for publication in J. I had a few comments as follows:
1) Introduction: the last paragraph of the Introduction section must give a brief idea about the research gap and the flow of research work. Also, the paragraphs in this section can be merged.
2) Page 2 Line nos. 82-84: please complement and enrich current statement with more related literatures: “Effects of the Surfactant, Polymer, and Crude Oil Properties on the Formation and Stabilization of Oil-Based Foam Liquid Films: Insights from the Microscale(2022). J Molecular Liquids, 373, 121194”; “Modeling of kinetic characteristics of alkaline‑surfactant-polymer-strengthened foams decay under ultrasonic standing wave(2022). Petroleum Science”.
3) Foaming capacity measurement: could you please present the structural schematic diagram of the apparatus?
4) The temperature for experiments is at 49 degree. How the temperature will affect the properties of
enhanced foam? Moreover, why is the surface tension test at 25 degree? Please clarify.
5) Page 6 Line nos. 206-208: The author should write necessary references. I do think the following paper can support the Ostwald ripening mechanisms: Foaming Properties and Foam Structure of Produced Liquid in Alkali/Surfactant/Polymer Flooding Production (2021).Journal of Energy Resources Technology, 143(10): 103005. Also, the Platonic effect can be discussed according to the Figure 3.
6) Figure 6: these two curves should be placed together.
7) Page 11 Line 331: the formula should be edited in separate row.
8) Please put enough emphasis on the points of novelty of the proposed study in your Conclusions.
9) Please indicate the physical meaning of the horizontal axis in Figures 2 & 5.
Author Response
Reviewer #2
Overall this paper is well structured and has the clean logic and necessary depth to be suitable for publication in J. I had a few comments as follows:
1) Introduction: the last paragraph of the Introduction section must give a brief idea about the research gap and the flow of research work. Also, the paragraphs in this section can be merged.
Reply: We thank the reviewer positive comments and suggestions. Following the reviewers’ suggestion we added a sentence about the research gap and re-arranged some sentences.
2) Page 2 Line nos. 82-84: please complement and enrich current statement with more related literatures: “Effects of the Surfactant, Polymer, and Crude Oil Properties on the Formation and Stabilization of Oil-Based Foam Liquid Films: Insights from the Microscale(2022). J Molecular Liquids, 373, 121194”; “Modeling of kinetic characteristics of alkaline‑surfactant-polymer-strengthened foams decay under ultrasonic standing wave(2022). Petroleum Science”.
Reply: We added the suggested references to the sentence.
3) Foaming capacity measurement: could you please present the structural schematic diagram of the apparatus?
Reply: A schema of the foam measuring apparatus was added to the manuscript (Figure 2).
4) The temperature for experiments is at 49 degree. How the temperature will affect the properties of enhanced foam? Moreover, why is the surface tension test at 25 degree? Please clarify.
Reply: We followed the standard ASTM D-1173-53 to measure the foam capacity of the formulations. In this standard, the tests are performed at 49 ºC. This work is a result of a collaboration with a company, and they are interested in testing the foam in a close way to the real application of some formulations, such as dishwashing detergents, which are used in a mid-high temperature, 49 ºC.
Studies indicate higher temperatures have an impact on the drainage and foam decay, resulting in lower stability of the formed foams.
The surface tension was measured at 25 ºC because is the more usual temperature of these determinations and we can easily compare it to other works from literature. It is known that an increase in temperature results in a decrease in surface tension. However, as the main interest of the company, for the desired applications, was related to the foam volume and stability, we decided to measure the foam volume at a temperature close to the application temperature and following a standard, but not all parameters.
5) Page 6 Line nos. 206-208: The author should write necessary references. I do think the following paper can support the Ostwald ripening mechanisms: Foaming Properties and Foam Structure of Produced Liquid in Alkali/Surfactant/Polymer Flooding Production (2021). Journal of Energy Resources Technology, 143(10): 103005. Also, the Platonic effect can be discussed according to the Figure 3.
Reply: We thank the reviewer's suggestions which have been included in the revised manuscript.
6) Figure 6: these two curves should be placed together.
Reply: We agree with the reviewer, and the curves were merged.
7) Page 11 Line 331: the formula should be edited in separate row.
Reply: The formula was edited in a separate row.
8) Please put enough emphasis on the points of novelty of the proposed study in your Conclusions.
Reply: We added some points to the conclusions emphasizing the novelty of the study.
9) Please indicate the physical meaning of the horizontal axis in Figures 2 & 5.
Reply: We apologize for the displacement of the horizontal axis in Figures 2 and 5. We have corrected the axes to indicate the meaning, which is the value of the surfactants alone.

Reviewer 3 Report
Comments and Suggestions for Authors
Reviewer’s comments on the manuscript: “Synergisms between surfactants, polymers and alcohols to improve the foamability of mixed systems” written by Luís Alves, Solange Magalhães, Cátia Esteves, Marco Sebastião and Filipe Antunes
The reviewed manuscript presents the effect of the addition of hydrophobically modified linear polymers, non-ionic co-surfactant and hydrotropes, and its mixtures, on the foam capacity of an anionic surfactant (alcohols, C12-14, ethoxylated, sulphates, sodium salts (2 EO) - SLE2S) and a zwitterionic surfactant (cocamidopropyl betaine – CAPB) solutions were evaluated through its foamability (foam formation and foam stability) and tensiometry. In my opinion the manuscript is in the journal’s fields of interests. Moreover, it is interesting and well-written. Experiments are properly planned and the obtained data are clear. Presented discussion is also convincing. I recommend to accept this paper after minor revisions.
Things that should be improved/added before the publication:
- Definitions/Abstract: To be honest, this is the first time I see a Definition instead of an Abstract at the beginning of the article and I am not convinced of the accuracy of this idea. This text resamples rather the Introduction than the Abstract. Why did the authors decide on this solution? I would prefer to read a simple abstract clearly presenting the purpose of the research, the experiments performed, the most important conclusions and the significance of the research.
- lines 29-30: “Foam is a colloidal dispersion, in which a gas is dispersed in a continuous liquid phase [1]”. From colloidal point of view his definition of foam is not correct because in the case of foams we can have liquid or solid as continuous phases.
- lines 36-37: Stable foams do not present spherical bubbles but foam cells, polyhedral, separated by flat liquid films. I think this is too much of a generalization. It is true that polyhedral foams are more durable, but even the stable beer foam mentioned by the authors above will contain both spherical and polyhedral bubbles, depending on the height and time. Please take this into account.
- Entire Materials and methods: The names of chemical substances in English should be written in lower case, not upper case. Please add information about the type of percentages (weight-in-weight (w/w), volume-in-volume (v/v) or weight-in-volume (w/v).
- Fig. 1: The quality of the figure is nor satisfactory.
- Editorial mistakes, lines: 87 (one extra space), 112 (one extra space), 148, 156, 245 (unnecessary underlining), 249, 323, 327 (units)
- Entire manuscript: Please decide if you want to make space between number and its unit and do it consequently.
- Entire manuscript: Please try to placed a number and its unit in the same line.
- Figs. 2, 4, 5: Please add info about uncertainty of measurements.
- lines 305-306 and 331: Please present all chemical equations in a separate lines with the explanation of each symbol.
- Fig. 7: Add the caption. I like this informative scheme bit its quality should be better.
Author Response
Reviewer #3
Reviewer’s comments on the manuscript: “Synergisms between surfactants, polymers and alcohols to improve the foamability of mixed systems” written by Luís Alves, Solange Magalhães, Cátia Esteves, Marco Sebastião and Filipe Antunes
The reviewed manuscript presents the effect of the addition of hydrophobically modified linear polymers, non-ionic co-surfactant and hydrotropes, and its mixtures, on the foam capacity of an anionic surfactant (alcohols, C12-14, ethoxylated, sulphates, sodium salts (2 EO) - SLE2S) and a zwitterionic surfactant (cocamidopropyl betaine – CAPB) solutions were evaluated through its foamability (foam formation and foam stability) and tensiometry. In my opinion the manuscript is in the journal’s fields of interests. Moreover, it is interesting and well-written. Experiments are properly planned and the obtained data are clear. Presented discussion is also convincing. I recommend to accept this paper after minor revisions.
Reply: We thank the reviewer for the positive comments on our manuscript.
Things that should be improved/added before the publication:
- Definitions/Abstract: To be honest, this is the first time I see a Definition instead of an Abstract at the beginning of the article and I am not convinced of the accuracy of this idea. This text resamples rather the Introduction than the Abstract. Why did the authors decide on this solution? I would prefer to read a simple abstract clearly presenting the purpose of the research, the experiments performed, the most important conclusions and the significance of the research.
Reply: We apologize for the misunderstanding. In fact, the manuscript was first submitted to another journal (Encyclopedia MDPI), which uses this designation, and as the manuscript was transferred to J (by suggestion of the Editorial team) the “Definition” terms persist in the version to journal J. We corrected the term to “Abstract” and revised the abstract text.
- lines 29-30: “Foam is a colloidal dispersion, in which a gas is dispersed in a continuous liquid phase [1]”. From colloidal point of view his definition of foam is not correct because in the case of foams we can have liquid or solid as continuous phases.
Reply: We thank the reviewer's comment. We have corrected the definition.
- lines 36-37: Stable foams do not present spherical bubbles but foam cells, polyhedral, separated by flat liquid films. I think this is too much of a generalization. It is true that polyhedral foams are more durable, but even the stable beer foam mentioned by the authors above will contain both spherical and polyhedral bubbles, depending on the height and time. Please take this into account.
Reply: We thank the relevant comment. We have in consideration the comments and we revised the text.
- Entire Materials and methods: The names of chemical substances in English should be written in lower case, not upper case. Please add information about the type of percentages (weight-in-weight (w/w), volume-in-volume (v/v) or weight-in-volume (w/v).
Reply: We corrected the names of the chemicals and added the information about percentages.
- Fig. 1: The quality of the figure is nor satisfactory.
Reply: We improved the image quality.
- Editorial mistakes, lines: 87 (one extra space), 112 (one extra space), 148, 156, 245 (unnecessary underlining), 249, 323, 327 (units)
Reply: We solved the editorial mistakes. We thank the reviewer for the careful analysis of the text.
- Entire manuscript: Please decide if you want to make space between number and its unit and do it consequently.
Reply: We become coherent along with the manuscript. We have chosen to add space.
- Entire manuscript: Please try to placed a number and its unit in the same line.
Reply: We tried to follow the reviewer's suggestion. We do our best to keep the number and its unit in the same line.
- Figs. 2, 4, 5: Please add info about uncertainty of measurements.
Reply: We thank the reviewer’s comment. Indeed the presented the foam volume and surface tension are presented as the average values. We apologize by dot not presenting the standard deviation. Now, we added the standard deviation to the data of foam volume and surface tension.
- lines 305-306 and 331: Please present all chemical equations in a separate lines with the explanation of each symbol.
Reply: We follow the suggestion, and the equations are presented in separate lines.
- Fig. 7: Add the caption. I like this informative scheme bit its quality should be better.
Reply: We thank the reviewer positive comment. We improved the image quality accordingly. Additionally, we uploaded a high-quality image to the journal webpage, because sometimes when included in a Word document the images lose quality.

Round 2
Reviewer 1 Report
Comments and Suggestions for Authors
The authors took into account the reviewer’s comments and made corrections to the new version of the article. The revised version may be recommended for publication